# HIERARCHICAL MEMORY NETWORKS

**Sarath Chandar**[*,1]**, Sungjin Ahn**[1]**, Hugo Larochelle**[2,4]**, Pascal Vincent**[1,4]**,
Gerald Tesauro**[3]**, Yoshua Bengio**[1,4]

[1] Université de Montréal, Canada.
[2] Twitter, USA.
[3] IBM Watson Research Center, USA.
[4] CIFAR, Canada.

## ABSTRACT

Memory networks are neural networks with an explicit memory component that can be both read and written to by the network. The memory is often addressed in a soft way using a softmax function, making end-to-end training with backpropagation possible. However, this is not computationally scalable for applications which require the network to read from extremely large memories. On the other hand, it is well known that hard attention mechanisms based on reinforcement learning are challenging to train successfully. In this paper, we explore a form of hierarchical memory network, which can be considered as a hybrid between hard and soft attention memory networks. The memory is organized in a hierarchical structure such that reading from it is done with less computation than soft attention over a flat memory, while also being easier to train than hard attention over a flat memory. Specifically, we propose to incorporate Maximum Inner Product Search (MIPS) in the training and inference procedures for our hierarchical memory network. We explore the use of various state-of-the art approximate MIPS techniques and report results on SimpleQuestions, a challenging large scale factoid question answering task.

## 1 INTRODUCTION

Until recently, traditional machine learning approaches for challenging tasks such as image captioning, object detection, or machine translation have consisted in complex pipelines of algorithms, each being separately tuned for better performance. With the recent success of neural networks and deep learning research, it has now become possible to train a single model end-to-end, using backpropagation. Such end-to-end systems often outperform traditional approaches, since the entire model is directly optimized with respect to the final task at hand. However, simple encode-decode style neural networks often underperform on knowledge-based reasoning tasks like question-answering or dialog systems. Indeed, in such cases it is nearly impossible for regular neural networks to store all the necessary knowledge in their parameters.

Neural networks with memory (Graves et al., 2014; Weston et al., 2015b) can deal with knowledge bases by having an external memory component which can be used to explicitly store knowledge. The memory is accessed by reader and writer functions, which are both made differentiable so that the entire architecture (neural network, reader, writer and memory components) can be trained end-to-end using backpropagation. Memory-based architectures can also be considered as generalizations of RNNs and LSTMs, where the memory is analogous to recurrent hidden states. However they are much richer in structure and can handle very long-term dependencies because once a vector (i.e., a memory) is stored, it is copied from time step to time step and can thus stay there for a very long time (and gradients correspondingly flow back time unhampered).

There exists several variants of neural networks with a memory component: Memory Networks (Weston et al., 2015b), Neural Turing Machines (NTM) (Graves et al., 2014), Dynamic Memory Net-

---

*Corresponding author: apsarathchandar@gmail.com

works (DMN) (Kumar et al., 2015). They all share five major components: memory, input module, reader, writer, and output module.

**Memory:** The memory is an array of cells, each capable of storing a vector. The memory is often initialized with external data (e.g. a database of facts), by filling in its cells with a pre-trained vector representations of that data.

**Input module:** The input module is to compute a representation of the input that can be used by other modules.

**Writer:** The writer takes the input representation and updates the memory based on it. The writer can be as simple as filling the slots in the memory with input vectors in a sequential way (as often done in memory networks). If the memory is bounded, instead of sequential writing, the writer has to decide where to write and when to rewrite cells (as often done in NTMs).

**Reader:** Given an input and the current state of the memory, the reader retrieves content from the memory, which will then be used by an output module. This often requires comparing the input's representation or a function of the recurrent state with memory cells using some scoring function such as a dot product.

**Output module:** Given the content retrieved by the reader, the output module generates a prediction, which often takes the form of a conditional distribution over multiple labels for the output.

For the rest of the paper, we will use the name *memory network* to describe any model which has any form of these five components. We would like to highlight that all the components except the memory are learnable. Depending on the application, any of these components can also be fixed. In this paper, we will focus on the situation where a network does not write and only reads from the memory.

In this paper, we focus on the application of memory networks to large-scale tasks. Specifically, we focus on large scale factoid question answering. For this problem, given a large set of facts and a natural language question, the goal of the system is to answer the question by retrieving the supporting fact for that question, from which the answer can be derived. Application of memory networks to this task has been studied by Bordes et al. (2015). However, Bordes et al. (2015) depended on keyword based heuristics to filter the facts to a smaller set which is manageable for training. However heuristics are invariably dataset dependent and we are interested in a more general solution which can be used when the facts are of any structure. One can design soft attention retrieval mechanisms, where a convex combination of all the cells is retrieved or design hard attention retrieval mechanisms where one or few cells from the memory are retrieved. Soft attention is achieved by using softmax over the memory which makes the reader differentiable and hence learning can be done using gradient descent. Hard attention is achieved by using methods like REINFORCE (Williams, 1992), which provides a noisy gradient estimate when discrete stochastic decisions are made by a model.

Both soft attention and hard attention have limitations. As the size of the memory grows, soft attention using softmax weighting is not scalable. It is computationally very expensive, since its complexity is linear in the size of the memory. Also, at initialization, gradients are dispersed so much that it can reduce the effectiveness of gradient descent. These problems can be alleviated by a hard attention mechanism, for which the training method of choice is REINFORCE. However, REINFORCE can be brittle due to its high variance and existing variance reduction techniques are complex. Thus, it is rarely used in memory networks (even in cases of a small memory).

In this paper, we propose a new memory selection mechanism based on Maximum Inner Product Search (MIPS) which is both scalable and easy to train. This can be considered as a hybrid of soft and hard attention mechanisms. The key idea is to structure the memory in a hierarchical way such that it is easy to perform MIPS, hence the name Hierarchical Memory Network (HMN). HMNs are scalable at both training and inference time. The main contributions of the paper are as follows:

- We explore hierarchical memory networks, where the memory is organized in a hierarchical fashion, which allows the reader to efficiently access only a subset of the memory.

- While there are several ways to decide which subset to access, we propose to pose memory access as a maximum inner product search (MIPS) problem.

- We empirically show that exact MIPS-based algorithms not only enjoy similar convergence as soft attention models, but can even improve the performance of the memory network.

- Since exact MIPS is as computationally expensive as a full soft attention model, we propose to train the memory networks using approximate MIPS techniques for scalable memory access.

- We empirically show that unlike exact MIPS, approximate MIPS algorithms provide a speedup and scalability of training, though at the cost of some performance.

## 2 HIERARCHICAL MEMORY NETWORKS

In this section, we describe the proposed Hierarchical Memory Network (HMN). In this paper, HMNs only differ from regular memory networks in two of its components: the memory and the reader.

**Memory:** Instead of a flat array of cells for the memory structure, HMNs leverages a hierarchical memory structure. Memory cells are organized into groups and the groups can further be organized into higher level groups. The choice for the memory structure is tightly coupled with the choice of reader, which is essential for fast memory access. We consider three classes of approaches for the memory's structure: hashing-based approaches, tree-based approaches, and clustering-based approaches. This is explained in detail in the next section.

**Reader:** The reader in the HMN is different from the readers in flat memory networks. Flat memory-based readers use either soft attention over the entire memory or hard attention that retrieves a single cell. While these mechanisms might work with small memories, with HMNs we are more interested in achieving scalability towards very large memories. So instead, HMN readers use soft attention only over a selected subset of the memory. Selecting memory subsets is guided by a maximum inner product search algorithm, which can exploit the hierarchical structure of the organized memory to retrieve the most relevant facts in sub-linear time. The MIPS-based reader is explained in more detail in the next section.

In HMNs, the reader is thus trained to create MIPS queries such that it can retrieve a sufficient set of facts. While most of the standard applications of MIPS (Ram & Gray, 2012; Bachrach et al., 2014; Shrivastava & Li, 2014) so far have focused on settings where both query vector and database (memory) vectors are precomputed and fixed, memory readers in HMNs are learning to do MIPS by updating the input representation such that the result of MIPS retrieval contains the correct fact(s).

## 3 MEMORY READER WITH $K$-MIPS ATTENTION

In this section, we describe how the HMN memory reader uses Maximum Inner Product Search (MIPS) during learning and inference.

We begin with a formal definition of $K$-MIPS. Given a set of points $\mathcal{X} = \{x_1, \ldots, x_n\}$ and a query vector $q$, our goal is to find

$$\operatorname{argmax}_{i \in \mathcal{X}}^{(K)} \ q^\top x_i \tag{1}$$

where the $\operatorname{argmax}^{(K)}$ returns the indices of the top-$K$ maximum values. In the case of HMNs, $\mathcal{X}$ corresponds to the memory and $q$ corresponds to the vector computed by the input module.

A simple but inefficient solution for $K$-MIPS involves a linear search over the cells in memory by performing the dot product of $q$ with all the memory cells. While this will return the exact result for $K$-MIPS, it is too costly to perform when we deal with a large-scale memory. However, in many practical applications, it is often sufficient to have an approximate result for $K$-MIPS, trading speed-up at the cost of the accuracy. There exist several approximate $K$-MIPS solutions in the literature (Shrivastava & Li, 2014; 2015; Bachrach et al., 2014; Neyshabur & Srebro, 2015).

All the approximate $K$-MIPS solutions add a form of hierarchical structure to the memory and visit only a subset of the memory cells to find the maximum inner product for a given query. Hashing-based approaches (Shrivastava & Li, 2014; 2015; Neyshabur & Srebro, 2015) hash cells into multiple bins, and given a query they search for $K$-MIPS cell vectors only in bins that are close to the bin

associated with the query. Tree-based approaches (Ram & Gray, 2012; Bachrach et al., 2014) create search trees with cells in the leaves of the tree. Given a query, a path in the tree is followed and MIPS is performed only for the leaf for the chosen path. Clustering-based approaches (Auvolat et al., 2015) cluster cells into multiple clusters (or a hierarchy of clusters) and given a query, they perform MIPS on the centroids of the top few clusters. We refer the readers to (Auvolat et al., 2015) for an extensive comparison of various state-of-the-art approaches for approximate $K$-MIPS.

Our proposal is to exploit this rich approximate $K$-MIPS literature to achieve scalable training and inference in HMNs. Instead of filtering the memory with heuristics, we propose to organize the memory based on approximate $K$-MIPS algorithms and then train the reader to learn to perform MIPS. Specifically, consider the following softmax over the memory which the reader has to perform for every reading step to retrieve a set of relevant candidates:

$$R_{out} = \text{softmax}(h(q)M^T) \qquad (2)$$

where $h(q) \in \mathbb{R}^d$ is the representation of the query, $M \in \mathbb{R}^{N \times d}$ is the memory with $N$ being the total number of cells in the memory. We propose to replace this softmax with $\text{softmax}^{(K)}$ which is defined as follows:

$$C = \text{argmax}^{(K)} \ h(q)M^T \qquad (3)$$

$$R_{out} = \text{softmax}^{(K)}(h(q)M^T) = \text{softmax}(h(q)M[C]^T) \qquad (4)$$

where $C$ is the indices of top-$K$ MIP candidate cells and $M[C]$ is a sub-matrix of $M$ where the rows are indexed by $C$.

One advantage of using the $\text{softmax}^{(K)}$ is that it naturally focuses on cells that would normally receive the strongest gradients during learning. That is, in a full softmax, the gradients are otherwise more dispersed across cells, given the large number of cells and despite many contributing a small gradient. As our experiments will show, this results in slower training.

One problematic situation when learning with the $\text{softmax}^{(K)}$ is when we are at the initial stages of training and the $K$-MIPS reader is not including the correct fact candidate. To avoid this issue, we always include the correct candidate to the top-$K$ candidates retrieved by the $K$-MIPS algorithm, effectively performing a fully supervised form of learning.

During training, the reader is updated by backpropagation from the output module, through the subset of memory cells. Additionally, the log-likelihood of the correct fact computed using $K$-softmax is also maximized. This second supervision helps the reader learn to modify the query such that the maximum inner product of the query with respect to the memory will yield the correct supporting fact in the top $K$ candidate set.

Until now, we described the exact $K$-MIPS-based learning framework, which still requires a linear look-up over all memory cells and would be prohibitive for large-scale memories. In such scenarios, we can replace the exact $K$-MIPS in the training procedure with the approximate $K$-MIPS. This is achieved by deploying a suitable memory hierarchical structure. The same approximate $K$-MIPS-based reader can be used during inference stage as well. Of course, approximate $K$-MIPS algorithms might not return the exact MIPS candidates and will likely to hurt performance, but at the benefit of achieving scalability.

While the memory representation is fixed in this paper, updating the memory along with the query representation should improve the likelihood of choosing the correct fact. However, updating the memory will reduce the precision of the approximate $K$-MIPS algorithms, since all of them assume that the vectors in the memory are static. Designing efficient dynamic $K$-MIPS should improve the performance of HMNs even further, a challenge that we hope to address in future work.

## 3.1 READER WITH CLUSTERING-BASED APPROXIMATE $K$-MIPS

Clustering-based approximate $K$-MIPS was proposed in (Auvolat et al., 2015) and it has been shown to outperform various other state-of-the-art data dependent and data independent approximate $K$-MIPS approaches for inference tasks. As we will show in the experiments section, clustering-based MIPS also performs better when used to training HMNs. Hence, we focus our presentation on the clustering-based approach and propose changes that were found to be helpful for learning HMNs.

Following most of the other approximate $K$-MIPS algorithms, Auvolat et al. (2015) convert MIPS to Maximum Cosine Similarity Search (MCSS) problem:

$$\text{argmax}_{i\in\mathcal{X}}^{(K)} \frac{q^T x_i}{||q||\ ||x_i||} = \text{argmax}_{i\in\mathcal{X}}^{(K)} \frac{q^T x_i}{||x_i||} \tag{5}$$

When all the data vectors $x_i$ have the same norm, then MCSS is equivalent to MIPS. However, it is often restrictive to have this additional constraint. Instead, Auvolat et al. (2015) append additional dimensions to both query and data vectors to convert MIPS to MCSS. In HMN terminology, this would correspond to adding a few more dimensions to the memory cells and input representations.

The algorithm introduces two hyper-parameters, $U < 1$ and $m \in \mathbb{N}^*$. The first step is to scale all the vectors in the memory by the same factor, such that $\max_i ||x_i||_2 = U$. We then apply two mappings, $P$ and $Q$, on the memory cells and on the input vector, respectively. These two mappings simply concatenate $m$ new components to the vectors and make the norms of the data points all roughly the same (Shrivastava & Li, 2015). The mappings are defined as follows:

$$P(x) = [x, 1/2 - ||x||_2^2, 1/2 - ||x||_2^4, \ldots, 1/2 - ||x||_2^{2^m}] \tag{6}$$
$$Q(x) = [x, 0, 0, \ldots, 0] \tag{7}$$

We thus have the following approximation of MIPS by MCSS for any query vector $q$:

$$\text{argmax}_i^{(K)} q^\top x_i \simeq \text{argmax}_i^{(K)} \frac{Q(q)^\top P(x_i)}{||Q(q)||_2 \cdot ||P(x_i)||_2} \tag{8}$$

Once we convert MIPS to MCSS, we can use spherical $K$-means (Zhong, 2005) or its hierarchical version to approximate and speedup the cosine similarity search. Once the memory is clustered, then every read operation requires only $K$ dot-products, where $K$ is the number of cluster centroids.

Since this is an approximation, it is error-prone. As we are using this approximation for the learning process, this introduces some bias in gradients, which can affect the overall performance of HMN. To alleviate this bias, we propose three simple strategies.

- Instead of using only the top-$K$ candidates for a single read query, we also add top-$K$ candidates retrieved for every other read query in the mini-batch. This serves two purposes. First, we can do efficient matrix multiplications by leveraging GPUs since all the $K$-softmax in a minibatch are over the same set of elements. Second, this also helps to decrease the bias introduced by the approximation error.
- For every read access, instead of only using the top few clusters which has a maximum product with the read query, we also sample some clusters from the rest, based on a probability distribution log-proportional to the dot product with the cluster centroids. This also decreases the bias.
- We can also sample random blocks of memory and add it to top-$K$ candidates.

We empirically investigate the effect of these variations in Section 5.5.

## 4 RELATED WORK

Memory networks have been introduced in (Weston et al., 2015b) and have been so far applied to comprehension-based question answering (Weston et al., 2015a; Sukhbaatar et al., 2015), large scale question answering (Bordes et al., 2015) and dialogue systems (Dodge et al., 2015). While (Weston et al., 2015b) considered supervised memory networks in which the correct supporting fact is given during the training stage, (Sukhbaatar et al., 2015) introduced semi-supervised memory networks that can learn the supporting fact by itself. (Kumar et al., 2015; Xiong et al., 2016) introduced Dynamic Memory Networks (DMNs) which can be considered as a memory network with two types of memory: a regular large memory and an episodic memory. Another related class of model is the Neural Turing Machine (Graves et al., 2014), which uses softmax-based soft attention. Later (Zaremba & Sutskever, 2015) extended NTM to hard attention using reinforcement learning. (Dodge et al., 2015; Bordes et al., 2015) alleviate the problem of the scalability of soft attention by having

an initial keyword based filtering stage, which reduces the number of facts being considered. Our work generalizes this filtering by using MIPS for filtering. This is desirable because MIPS can be applied for any modality of data or even when there is no overlap between the words in a question and the words in facts.

The softmax arises in various situations and most relevant to this work are scaling methods for large vocabulary neural language modeling. In neural language modeling, the final layer is a softmax distribution over the next word and there exist several approaches to achieve scalability. (Morin & Bengio, 2005) proposes a hierarchical softmax based on prior clustering of the words into a binary, or more generally $n$-ary tree, that serves as a fixed structure for the learning process of the model. The complexity of training is reduced from $O(n)$ to $O(\log n)$. Due to its clustering and tree structure, it resembles the clustering-based MIPS techniques we explore in this paper. However, the approaches differ at a fundamental level. Hierarchical softmax defines the probability of a leaf node as the product of all the probabilities computed by all the intermediate softmaxes on the way to that leaf node. By contrast, an approximate MIPS search imposes no such constraining structure on the probabilistic model, and is better thought as efficiently searching for top winners of what amounts to be a large ordinary flat softmax. Other methods such as Noice Constrastive Estimation (Mnih & Gregor, 2014) and Negative Sampling (Mikolov et al., 2013) avoid an expensive normalization constant by sampling negative samples from some marginal distribution. By contrast, our approach approximates the softmax by explicitly including in its negative samples candidates that likely would have a large softmax value. Jean et al. (2015) introduces an importance sampling approach that considers all the words in a mini-batch as the candidate set. This in general might also not include the MIPS candidates with highest softmax values.

(Spring & Shrivastava, 2016) is the only work that we know of, proposing to use MIPS during learning. It proposes hashing-based MIPS to sort the hidden layer activations and reduce the computation in every layer. However, a small scale application was considered and data-independent methods like hashing will likely suffer as dimensionality increases. Rae et al. (2016) have also independently proposed a model called SAM to use approximate search methods for memory access in NTM-like architectures. However, our motivation is different. While Rae et al. (2016) focus on architectures where the memory is written by the controller itself, we focus on handling memory access to large external knowledge bases. While both the models fix the memory access mechanism (HMN uses MIPS and SAM uses NNS), our controller works in a much more constrained setting. Moreover, our experiments suggest that the performance of SAM could be improved using a clustering-based approach as in our work, instead of tree/hash-based approaches for memory search used by SAM.

## 5 EXPERIMENTS

In this section, we report experiments on factoid question answering using hierarchical memory networks. Specifically, we use the SimpleQuestions dataset Bordes et al. (2015). The aim of these experiments is not to achieve state-of-the-art results on this dataset. Rather, we aim to propose and analyze various approaches to make memory networks more scalable and explore the achieved tradeoffs between speed and accuracy.

### 5.1 DATASET

We use SimpleQuestions (Bordes et al., 2015) which is a large scale factoid question answering dataset. SimpleQuestions consists of 108,442 natural language questions, each paired with a corresponding fact from Freebase. Each fact is a triple (subject,relation,object) and the answer to the question is always the object. The dataset is divided into training (75910), validation (10845), and test (21687) sets. Unlike Bordes et al. (2015) who additionally considered FB2M (10M facts) or FB5M (12M facts) with keyword-based heuristics for filtering most of the facts for each question, we only use SimpleQuestions, with no keyword-based heuristics. This allows us to do a direct comparison with the full softmax approach in a reasonable amount of time. Moreover, we would like to highlight that for this dataset, keyword-based filtering is a very efficient heuristic since all questions have an appropriate source entity with a matching word. Nevertheless, our goal is to design a general purpose architecture without such strong assumptions on the nature of the data.

## 5.2 MODEL

Let $V_q$ be the vocabulary of all words in the natural language questions. Let $W_q$ be a $|V_q| * m$ matrix where each row is some $m$ dimensional embedding for a word in the question vocabulary. This matrix is initialized with random values and learned during training. Given any question, we represent it with a bag-of-words representation by summing the vector representation of each word in the question. Let $q = \{w_i\}_{i=1}^p$,

$$h(q) = \sum_{i=1}^{p} W_q[w_i]$$

Then, to find the relevant fact from the memory M, we call the $K$-MIPS-based reader module with $h(q)$ as the query. This uses Equation 3 and 4 to compute the output of the reader $R_{out}$. The reader is trained by minimizing the Negative Log Likelihood (NLL) of the correct fact.

$$\mathcal{J}_\theta = \sum_{i=1}^{N} -\log(R_{out}[f_i])$$

where $f_i$ is the index of the correct fact in $W_m$. We are fixing the memory embeddings to the TransE (Bordes et al., 2013) embeddings and learning only the question embeddings.

This model is simpler than the one reported in (Bordes et al., 2015) so that it is esay to analyze the effect of various memory reading strategies.

## 5.3 TRAINING DETAILS

We trained the model with the Adam optimizer (Kingma & Ba, 2014), with a fixed learning rate of 0.001. We used mini-batches of size 128. We used 200 dimensional embeddings for the TransE entities, yielding 600 dimensional embeddings for facts by concatenating the embeddings of the subject, relation and object. We also experimented with summing the entities in the triple instead of concatenating, but we found that it was difficult for the model to differentiate facts this way. The only learnable parameters by the HMN model are the question word embeddings. The entity distribution in SimpleQuestions is extremely sparse and hence, following Bordes et al. (2015), we also add artificial questions for all the facts for which we do not have natural language questions. Unlike Bordes et al. (2015), we do not add any other additional tasks like paraphrase detection to the model, mainly to study the effect of the reader. We stopped training for all the models when the validation accuracy consistently decreased for 3 epochs.

## 5.4 EXACT $K$-MIPS IMPROVES ACCURACY

In this section, we compare the performance of the full soft attention reader and exact $K$-MIPS attention readers. Our goal is to verify that $K$-MIPS attention is in fact a valid and useful attention mechanism and see how it fares when compared to full soft attention. For $K$-MIPS attention, we tried $K \in 10, 50, 100, 1000$. We would like to emphasize that, at training time, along with $K$ candidates for a particular question, we also add the $K$-candidates for each question in the mini-batch. So the exact size of the softmax layer would be higer than $K$ during training. In Table 1, we report the test performance of memory networks using the soft attention reader and $K$-MIPS attention reader. We also report the average softmax size during training. From the table, it is clear that the $K$-MIPS attention readers improve the performance of the network compared to soft attention reader. In fact, smaller the value of $K$ is, better the performance. This result suggests that it is better to use a $K$-MIPS layer instead of softmax layer whenever possible. It is interesting to see that the convergence of the model is not slowed down due to this change in softmax computation (as shown in Figure 1).

This experiment confirms the usefulness of $K$-MIPS attention. However, exact $K$-MIPS has the same complexity as a full softmax. Hence, to scale up the training, we need more efficient forms of $K$-MIPS attention, which is the focus of next experiment.

| Model | Test Acc. | Avg. Softmax Size |
|---|---|---|
| Full-softmax | 59.5 | 108442 |
| 10-MIPS | **62.2** | **1290** |
| 50-MIPS | 61.2 | 6180 |
| 100-MIPS | 60.6 | 11928 |
| 1000-MIPS | 59.6 | 70941 |
| Clustering | 51.5 | 20006 |
| PCA-Tree | 32.4 | 21108 |
| WTA-Hash | 40.2 | 20008 |

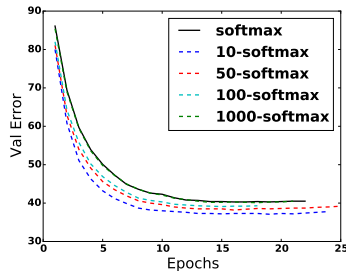

Table 1: Accuracy in SQ test-set and average size of memory used. 10-softmax has high performance while using only smaller amount of memory.

Figure 1: Validation curve for various models. Convergence is not slowed down by k-softmax.

## 5.5 Approximate $K$-MIPS based learning

As mentioned previously, designing faster algorithms for $K$-MIPS is an active area of research. Auvolat et al. (2015) compared several state-of-the-art data-dependent and data-independent methods for faster approximate $K$-MIPS and it was found that clustering-based MIPS performs significantly better than other approaches. However the focus of the comparison was on performance during the inference stage. In HMNs, $K$-MIPS must be used at both training stage and inference stages. To verify if the same trend can been seen during learning stage as well, we compared three different approaches:

**Clustering:** This was explained in detail in section 3.

**WTA-Hash:** Winner Takes All hashing (Vijayanarasimhan et al., 2014) is a hashing-based $K$-MIPS algorithm which also converts MIPS to MCSS by augmenting additional dimensions to the vectors. This method used $n$ hash functions and each hash function does $p$ different random permutations of the vector. Then the prefix constituted by the first $k$ elements of each permuted vector is used to construct the hash for the vector.

**PCA-Tree:** PCA-Tree (Bachrach et al., 2014) is the state-of-the-art tree-based method, which converts MIPS to NNS by vector augmentation. It uses the principal components of the data to construct a balanced binary tree with data residing in the leaves.

For a fair comparison, we varied the hyper-parameters of each algorithm in such a way that the average speedup is approximately the same. Table 1 shows the performance of all three methods, compared to a full softmax. From the table, it is clear that the clustering-based method performs significantly better than the other two methods. However, performances are lower when compared to the performance of the full softmax.

As a next experiment, we analyze various the strategies proposed in Section 3.1 to reduce the approximation bias of clustering-based $K$-MIPS:

**Top-K:** This strategy picks the vectors in the top $K$ clusters as candidates.

**Sample-K:** This strategy samples $K$ clusters, without replacement, based on a probability distribution based on the dot product of the query with the cluster centroids. When combined with the Top-$K$ strategy, we ignore clusters selected by the Top-$k$ strategy for sampling.

**Rand-block:** This strategy divides the memory into several blocks and uniformly samples a random block as candidate.

We experimented with 1000 clusters and 2000 clusters. While comparing various training strategies, we made sure that the effective speedup is approximately the same. Memory access to facts per query for all the models is approximately 20,000, hence yielding a 5X speedup.

Results are given in Table 2. We observe that the best approach is to combine the Top-K and Sample-K strategies, with Rand-block not being beneficial. Interestingly, the worst performances correspond to cases where the Sample-K strategy is ignored.

| Top-K | Sample-K | rand-block | 1000 clusters | | 2000 clusters | |
|---|---|---|---|---|---|---|
| | | | Test Acc. | epochs | Test Acc. | epochs |
| Yes | No | No | 50.2 | 16 | 51.5 | 22 |
| No | Yes | No | 52.5 | 68 | 52.8 | 63 |
| Yes | Yes | No | **52.8** | 31 | **53.1** | 26 |
| Yes | No | Yes | 51.8 | 32 | 52.3 | 26 |
| Yes | Yes | Yes | 52.5 | 38 | 52.7 | 19 |

Table 2: Accuracy in SQ test set and number of epochs for convergence.

## 6 CONCLUSION

In this paper, we proposed a hierarchical memory network that exploits $K$-MIPS for its attention-based reader. Unlike soft attention readers, $K$-MIPS attention reader is easily scalable to larger memories. This is achieved by organizing the memory in a hierarchical way. Experiments on the SimpleQuestions dataset demonstrate that exact $K$-MIPS attention is better than soft attention. However, existing state-of-the-art approximate $K$-MIPS techniques provide a speedup at the cost of some accuracy. Future research will investigate designing efficient dynamic $K$-MIPS algorithms, where the memory can be dynamically updated during training. This should reduce the approximation bias and hence improve the overall performance.

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
