# Peer review of "Hierarchical Memory Networks"

_ICLR 2017 — rejected_

[Official Review · AnonReviewer1 · rating 4 · confidence 4 · 17 Dec 2016]
**No Title**

I find this paper not very compelling.  The basic idea seems to be that we can put a fast neighbor searcher into a memory augmented net to make the memory lookups scalable.  However, this was precisely the point of Rae et al.    There are a  number of standardized neighbor searchers; I don't understand why the authors choose to use their own (which they do not benchmark against the standards).   Moreover, they test on a problem where there is no clear need for (vector based) fast-nn, because one can use hashing on the text.     I also find the repeated distinction between "mips" and "nns" distracting; most libraries that can do one can do the other, or inputs can be modified  to switch between the problems; indeed the authors do this when they convert to the  "mcss" problem.

[Official Review · AnonReviewer3 · rating 5 · confidence 5 · 19 Dec 2016]
**Not a very convincing proposal for dealing with very large memories in memory networks**

The paper proposes an algorithm for training memory networks which have very large memories. Training such models in traditional ways, by using soft-attention mechanism over all the memory slots is not only slow, it is also harder to train due to dispersion of gradients. The paper proposes to use the k-mips algorithm over the memories to choose a subset of the memory slots over which the attention is applied. Since the cost of exact k-mips is the same as doing full attention, the authors propose to use approximate k-mips, which while faster to compute, results in inferior performance. An artifact of using k-mips is that one cannot learn the memory slots. Hence they are pre-trained and kept fixed during entire training. The experimental section shows the efficacy of using k-mips using the SimpleQuestions dataset. The exact k-mips results in the same performance as the full attention. The approximate k-mips results in deterioration in performance. The paper is quite clearly written and easy to understand. 

I think the ideas proposed in the paper are not super convincing. I have a number of issues with this paper. 

1. The k-mips algorithm forces the memories to be fixed. This to me is a rather limiting constraint, especially on problems/dataset which will require multiple hops of training to do compounded reasoning. As a results I'm not entirely sure about the usefulness of this technique. 
2. Furthermore, the exact k-mips is the sample complexity as the full attention. The only way to achieve speedup is to use approx k-mips. That, as expected, results in a significant drop in performance. 
3. The paper motivates the ideas by proposing solutions to eliminate heuristics used to prune the memories. However in Section 3.1 the authors themselves end up using multiple heuristics to make the training work. Agreed, that the used heuristics are not data dependent, but still, it feels like they are kicking the can down the road as far as heuristics are concerned. 
4. The experimental results are not very convincing. First there is no speed comparison. Second, the authors do not compare with methods other than k-mips which do fast nearest neighbor search, such as, FLANN.

[Official Review · AnonReviewer2 · rating 5 · confidence 3 · 20 Dec 2016]
**No Title**

1. The hierarchical memory is fixed, not learned, and there is no hierarchical in the experimental section, only one layer for softmax layer.
2. It shows the 10-mips > 100-mips > 1000-mips, does it mean 1-mips is the best one we should adopt?
3. Approximated k-mips is worse than even original method. Why does it need exact k-mips? It seems the proposed method is not robust.

[Final Decision · Program Chairs · 06 Feb 2017]
**ICLR committee final decision**

This paper was reviewed by three experts. While they find interesting ideas in the manuscript, all three point to deficiencies (unconvincing results, etc) and unanimously recommend rejection.